# Negative Poisson’s Ratio-Spacer Design and Its Thermo-Mechanical Coupling Analysis Considering Specific Force Output

**DOI:** 10.3390/ma14123421

**Published:** 2021-06-21

**Authors:** Qianqian Yuan, Yongsheng Zhu, Ke Yan, Yiqing Cai, Jun Hong

**Affiliations:** Key Laboratory of Education Ministry for Modern Design and Rotor-Bearing System, Xi’an Jiaotong University, Xi’an 710049, China; y_qianqian@stu.xjtu.edu.cn (Q.Y.); yanke@mail.xjtu.edu.cn (K.Y.); caiyiqing8106@163.com (Y.C.); jhong@mail.xjtu.edu.cn (J.H.)

**Keywords:** negative Poisson’s ratio-spacer, low-porosity structure, rolling bearing, self-regulation of preload, thermo-mechanical coupling

## Abstract

Aiming at the problems of a complex structure or poor controllability of the existing bearing preload control devices, a method of self-regulation via a negative Poisson’s ratio (NPR) spacer is proposed. Firstly, the principle of preload automatic adjustment at the bearing operation was introduced and the NPRs with three types of cell structures were analyzed. Furthermore, a thermo-mechanical coupling analysis model of the NPR spacer was established and the deformation and force output characteristics of the NPR spacer were studied and experimentally verified. It is found that the concave hexagonal cell structure has the optimal deformation characteristics for bearing preload adjustment. When the temperature is considered, the absolute value of Poisson’s ratio of the NPR spacer decreases as the speed increases and the elongation of the NPR spacer and the output forces are much larger than those without temperature consideration. With the increase in temperature or rotating speed, the axial elongation and output forces of the NPR spacer increases while the effect of temperature is relatively larger.

## 1. Introduction

High-speed precision spindles are the core components of high-end machine tools. Compared with ordinary machine tool spindles, high-speed spindles have processing ranges that cover the entire process from low-speed roughing to high-speed finishing. Different processing stages have different requirements of preloading. Rough machining is a heavy cutting process and requires a high preload to ensure the stiffness of the spindle system and to resist the spindle deformation or the vibrations caused by the high cutting loads. In the finishing stage, the high speed and the low cutting load require an appropriate reduction in the preload to decrease the bearing heating and suppress the temperature rises. The traditional constant preload cannot fully meet the variable preload requirements of a high-speed spindle under a wide range of working conditions [1]. Therefore, it is important to study variable preloading technology for spindle bearings.

At present, mainly hydraulic, electromagnetic, and piezoelectric methods are being used worldwide in the studies on variable bearing preloads [2,3,4,5,6,7,8,9]. For example, Jiang and Choi designed an adjustment mechanism based on the hydraulic preload force [2,3]; however, the use of hydraulic technology requires a complex hydraulic system, which is expensive. Hwang et al. used electromagnetic loading technology to change the bearing preload [4]. Hu et al. designed a preload adjusting mechanism for spindle bearings based on a piezoelectric apparatus [5]. However, the electromagnetic force loading device is limited by the installation space, the preload adjustment range is small, and the piezoelectric actuator also has problems of easy aging and poor reliability. A preload adjustment method is an active adjustment approach, which requires a control system to relay a control command; therefore, it is difficult to realize self-adaptive preload adjustments within the full working speed range of high-speed spindles. Aiming at the problem of self-adaptive preload control, Kim et al. used a simple eccentric mass system to convert the centrifugal force into the axial force acting on a bearing to achieve automatic adjustments of the bearing preload force [6]. However, this preload force adaptive adjustment method changed the structure of spindle shaft systems and had low versatility. Therefore, in 2020, Kim et al. proposed another variable adjustment structure for spindle preload force using a Terfenol-D brake, which improves the response time and can adjust the spindle preload force within a certain range [7]. Razban et al. proposed a spring structure that can vary with speed to adjust the spindle preload force based on speed [8]; however, this method requires an additional structure. Yang et al. proposed to use the difference in the axial expansion caused by the different thermal expansion coefficients of the double-layer sleeve materials to adjust the bearing preload force [9]. Although this method achieves preload force adjustment without changing the structure of the shaft system, it relies on the temperature effect. Moreover, owing to the hysteresis of temperature change, it is difficult to realize the rapid response of the preload force to a change in the spindle operating speed and, moreover, its accuracy is difficult to control. For the above preload adjustment methods, either the device structure is complex, has low reliability, or is simple but with a slow response. This study designs a spacer based on an NPR structure to achieve adaptive adjustments of the bearing preload at different speeds.

An NPR structure is also called “auxetic” [10], i.e., when subjected to tension, it experiences lateral expansion in the elastic range and, when subjected to compression, it undergoes lateral contraction and its Poisson’s ratio is negative. Auxetic behavior is typically achieved by a distinct structural design consisting of periodically arranged auxetic units. These units can be classified into seven categories based on their structural features and deformation mechanisms: chiral structures, re-entrant structures, rotating rigid structures, origami-based and kirigami-based metamaterials, perforation structures, and auxetic foams [11]. Among them, chiral structures [12,13], re-entrant structures [14,15,16,17], and rotating rigid structures [17] are high-porosity structures used in seismic isolation, buffering, and noise reduction [17,18]. The origami-based and kirigami-based metamaterial patterns possess interesting mechanical features—one degree of freedom-mobility, auxetic in-plane behavior, and energy absorption capability—for applications such as core to sandwich structures, shock absorbers, and airless tires [19,20]. The above structure possesses high porosity, a high compression ratio, and low stiffness, which are unsuitable for achieving high capacities. Bearing spacers are subjected to certain loads in a spindle system and requires a certain stiffness and load capacity. Since an NPR structure is porous, its stiffness is less than that of a solid structure and high porosity implies low stiffness. In order to improve the load-carrying capacity and stiffness of an NPR structure, it is necessary to use a structure with low porosity and high stiffness. The porosity range of perforation structures can reach 2–8% by adjusting their structural parameters [20]. The research on low-porosity structures is mainly focused on perforation structures [21,22,23,24,25]. Taylor et al. found that thin plates with orthogonal elliptical voids exhibit auxetic behavior and that their Poisson’s ratio can be effectively tuned by adjusting the aspect ratio of an alternating pattern of elliptical voids, which demonstrates the ability to design an NPR [21]. Francesconi et al. studied the fatigue performance of positive Poisson’s ratio (PPR) geometries using circular holes compared to those of auxetic stop-hole and straight-groove hole geometries. The results showed that the fatigue life of a circular hole with a PPR is 20% lower than that of a straight slot hole with an NPR at the same porosity and peak effective maximum stress [23]. Chen et al. proposed a new topological configuration having low porosity and found that Poisson’s ratio and band gaps can be tuned effectively by tailoring the key geometric parameters of the perforation. Moreover, they analyzed the stress–strain behavior, tensile strength, and acoustic transmission characteristics of a low-porosity auxetic perforated mechanical metamaterials [24]. The above studies show that the properties of NPR materials can be effectively tuned by the design of NPR cells and their arrangement.

Concurrently, the aforementioned studies on low-porosity NPR structures were mainly aimed at two-dimensional planar structures. In this study, the aim was to realize real-time adjustment of the preload of angular contact ball bearings. Therefore, it was necessary to develop a three-dimensional (3D) annular NPR structure to achieve axial positioning and preload adjustments of the inner ring of the bearings (see Section 2 for a description of the principle). At present, the research on the design and application of negative Poisson’s ratio three-dimensional structures mainly focuses on high porosity [12,13,14,15,16] and the research on the design and application of low-porosity three-dimensional structures, especially on the deformation capacity and force output characteristics of such circular structures under centrifugal force and boundary constraints, has not been reported. Simultaneously, owing to the large variation in the internal temperature range of a spindle in the full operating range [25,26], the influence of temperature on the geometric and mechanical properties of an NPR spacer is nonnegligible. The method for the achievement of the determined output force while considering thermal coupling conditions is the core problem addressed in this study.

In view of the above, this study explores a low-porosity topological 3D NPR structure with high stiffness, considering thermo-mechanical coupling properties, and based on previous studies. The remainder of this paper is organized as follows: the introduction of the basic principle of the bearing preload adjustment and the determined boundary conditions of the NPR spacer force are presented in Section 2. The comparative analysis of several cell structures to preferably select the cell structure with the highest deformation capacity and the obtained 3D low-porosity annular NPR spacer using the periodic expansion of the cell structure are discussed in Section 3. The calculation method of the NPR spacer output force and the analytical model of the NPR spacer considering the thermal and centrifugal forces and axial constraints are presented in Section 4. In addition, the analysis of the force output characteristics of the NPR spacer and the comparison of the results with experimental ones are provided in Section 4. Finally, the conclusions are summarized in Section 5.

## 2. Principle of Adaptive Adjustment of Preload

Based on the requirement of optimal preload adjustment at full speed, a bearing inner ring spacer with an NPR is designed and installed on a spindle, as shown in Figure 1. The spindle-bearing system uses a positioning preload and the outer spacer is formed of conventional solid material with a PPR.

The principle of the NPR spacer to adjust the bearing preload is shown in Figure 2 In the figure, *F*_a_ is the initial preload of the bearing, *F* is the centrifugal force, *δ* is the axial elongation of the NPR spacer under centrifugal force and preload, and *F*_a_′ is the preload of the bearing after adjustment. In the initial assembly stage (Figure 2a), the Poisson’s ratio-controlled spacer structure is pressurized due to the bearing preload force *F*_a_ and the band gaps in each unit cell structure are eliminated due to the low porosity of the NPR spacer (Figure 2c). In this case, the structural stiffness of the NPR spacer is close to the stiffness of the base material, which ensures adequate stiffness of the shaft system at static or low speed operating conditions. During operation (Figure 2b), the NPR spacer is subjected to centrifugal force due to the velocity and so the cells expand in the direction of the vertical force (Figure 2e), which results in the expansion of the NPR spacer (Figure 2d). The NPR-controlled spacer structure slightly elongates in the axial direction and the extension direction is opposite to the bearing preload direction; therefore, the bearing preload decreases. When the speed decreases, the centrifugal force decreases and the axial force of the NPR spacer also decreases; moreover, the bearing shifts to the initial position under the action of the PPR spacer elastic force and the bearing preload force increases again.

Based on the bearing mechanical model, at rest, the relationship between the bearing axial load and the corresponding axial displacement is nonlinear. The specific calculation method is referred from [27] and only the simplified formula is provided as follows:(1)$$F_{a}=f\left(\delta_{a}\right)$$
where *δ*_a_ is the axial displacement of the bearing inner ring.

When working, the relationship between the axial load on the bearing and the displacement of the inner ring of the bearing is the following:(2)$${F^{\prime }}_{a}=f\left(\delta_{a}-\delta ,n\right)$$
where *n* is the bearing speed.

Thus, the NPR spacer output force *F*_af_ is
(3)$$F_{\mathrm{af}}=F_{a}-F_{a}{}^{\prime }$$

Based on the above working principle, the NPR spacer structure is designed to meet the initial stage of the large stiffness and also to ensure that the NPR spacer elongation in the range of the speed rise is close to or equal to the bearing displacement that reduces the bearing preload. In order to achieve controlled adjustment of the preload at different speeds, the NPR structure design considers different working conditions to achieve the ideal axial deformation at different speeds in order to realize the quantitative control of the preload. The large amount of heat generated during the. operation of the spindle system has a major impact on the spindle thermal displacement, rotation accuracy, and other performances [28]. The preload force is an important factor affecting the bearing heat generation. In order to ensure precise adjustment of the bearing preload, the proposed design of the spacer is a distinct 3D low-porosity spacer under the combined effect of the centrifugal force, heat, and the axial force.

## 3. Design of 3D NPR Spacer

The NPR structure is composed of a periodic arrangement of cells. In this study, first, the type of cell structure is determined, which is subsequently expanded to the NPR structure and repeating it at certain regular periods. Finally, the bearing NPR spacer is designed based on the deformation characteristics of the NPR structure.

### 3.1. Design of NPR Cell Structure

In this study, the low-porosity NPR structure is made up by a periodic arrangement of the unit cells, as shown in Figure 3. In the figure, dotted line denotes the NPR unit cell, the slit shape is the band gap, and *L*_0_ is the length of the unit cell.

Porosity is the percentage of pore volume in the bulk material compared to the total volume of the material in its natural state and the porosity is calculated as follows:(4)$$P=\frac{s}{L_{0}{}^{2}}\times 100\%$$
where *s* is the area of the solid in the unit cell.

Poisson’s ratio of the unit cell structure is 20 and described as follows:(5)$$\bar{\upsilon }=-\frac{{\langle u_{x}\rangle }^{R}-{\langle u_{x}\rangle }^{L}}{{\langle u_{y}\rangle }^{T}-{\langle u_{y}\rangle }^{B}}$$
where ${\langle u_{x}\rangle }^{L}$ and ${\langle u_{x}\rangle }^{R}$ are the average transverse displacements of the left and right boundaries of the representative volume element, respectively, and ${\langle u_{y}\rangle }^{T}$ and ${\langle u_{y}\rangle }^{B}$ are the average longitudinal displacements of the top and bottom boundaries of the representative volume element, respectively.

In order to ensure the stiffness of the shaft system, the bearing spacer is required to have sufficient stiffness and load-bearing capacities. At the same porosity, the NPR structure has the largest axial deformation, i.e., the absolute value of Poisson’s ratio of NPR structure is the largest. In this study, the Poisson’s ratios of three different unit cells with inner concave hexagonal, H-shaped, and star-shaped band gaps were calculated by the finite element method and their Poisson’s ratios were analyzed comparatively. The band gap structure parameters for the three types are listed in Table 1. Young’s modulus of the base material for each unit cell is 1.06 × 10^5^ MPa and the Poisson’s ratio of the base material for each unit cell is 0.324. Each unit cell is realized by the periodic topology with two band gaps vertically alternating, as shown in Figure 4 (the plate size is 240 mm × 40 mm) and the porosity of each unit cell is 8% calculated using Equation (4). The commercial finite element package, ANSYS, is used for all the simulations. Each mesh is constructed using a high-order 3D 20-node solid element (ANSYS element type Solid186) and the boundary condition is that the left end of the rectangular plate is completely fixed and a tension of 100 N is applied on the right end.

The contour plots of the horizontal (*u_x_*) and vertical (*u_y_*) components of the displacement field of the cell periodic structure are shown in Figure 5. In order to minimize the effect of the boundaries, the central part (40 mm × 40 mm, see the dashed red box in Figure 5) is selected to analyze the displacement of the unit cell structure and the calculation of Poisson’s ratio of the structure uses Equation (5). Poisson’s ratios of the three NPR unit cell structures—inner concave hexagonal, H-shaped, and Star-shaped—are −0.44, −0.0647, and −0.0354, respectively. At the same porosity, the unit cell with concave hexagonal band gaps has the highest absolute value of Poisson’s ratio and the deformation is the largest in the direction perpendicular to the force.

### 3.2. Three-Dimensional NPR Spacer Model

In the positioning of preload spindle-bearing system, the bearing preload is adjusted by the length difference between the inner and outer bearing spacers. The spindle system is supported by 7014C angular contact ball bearings. Based on the spindle structure geometry, the inner spacer of the bearings is designed with an inner diameter of 70 mm, the outer diameter of 76 mm, and axial length of 100 mm. Based on the results of the analysis discussed in Section 3.1, the concave hexagon is used as the cell of the NPR spacer structure. In view of the feature that the NPR structure can produce a two-way synchronous deformation when subjected to an in-plane tensile pressure, the cell is expanded periodically in the circumferential direction to form a toroidal spacer structure (Figure 6). Under the action of the radial centrifugal force, the circumferential deformation of the spacer increases which synchronously causes its axial elongation, generating an axial thrust on the bearing, and changing the preload of the bearing. The new structure has the characteristics of a simple structure and a wide range of output force (see Section 4).

## 4. Coupled Thermo-Mechanical Analysis of NPR Spacer

### 4.1. Calculation of Output Force of NPR Spacer

The output force of the NPR spacer is related to its rotational speed and bearing axial force and the procedure of calculating the output force of NPR spacer is shown in Figure 7.

### 4.2. Thermo-Mechanical Coupling Calculation of NPR Spacer

In order to obtain the deformation and force output characteristics of the NPR spacer under the combined effect of the thermal and force fields, its temperature distribution needs to be obtained first. The heat in a mechanical spindle system is mainly from friction between the bearing components and the temperature rise of the NPR spacer is mainly attributed to the heat transfer from the bearings. In order to obtain the temperature distributions of the NPR spacer at different speeds, we first calculate the bearing heat and subsequently construct a heat transfer model of the spindle-bearing system. This heat load is applied to the shaft system to obtain its temperature fields at different speeds (the calculation method of the spindle-bearing system temperature field can be referred from [29], which is not repeated here) from which the temperature distribution of the spacer is derived. The temperature parameters of the NPR spacer are imposed as boundary conditions in the NPR spacer mechanics model. The Poisson’s ratio value, elongation, and output forces of the NPR spacer are calculated under the combined action of temperature, centrifugal force, and axial force. The specific calculation process is shown in Figure 8.

### 4.3. Result Analysis

The thermo-mechanical coupling analysis of the NPR spacer is conducted based on a positioning and preloading spindle-bearing system. The bearing model is 7014C. The initial preload of the bearing is 1500 N. The ambient temperature is 30.1 °C. The spindle material is 38CrNoALA. The bearing material is GCr15. The finite element type is Solid186.

The calculated temperature distribution of the entire spindle-bearing system at the speed of 8000 r/min is shown in Figure 9 from which the temperature field distribution of the NPR spacer is extracted (Figure 10a). Using the same method, the obtained NPR spacer temperature fields at 2000 r/min, 4000 r/min, and 6000 r/min speeds of continuous operation of this spindle-bearing system reaches thermal equilibrium and the results are shown in Figure 10a–d. It can be observed that when the thermal equilibrium is reached, the temperature is high at both ends of the NPR spacer and lowest in the middle. Moreover, the temperature is close to the gradient distribution from one end to the middle. The temperature at the front end of the NPR spacer is slightly lower than that at the back end. The reason is that the front end of the spindle extends longer, which results in better heat dissipation. Thus, the front bearing temperature is also slightly lower than the rear bearing temperature, which eventually causes the NPR spacer temperature at the front section to be slightly lower than the rear temperature. The results of this NPR spacer temperature field calculation provide the data for imposing a temperature field on the NPR spacer in the subsequent thermo-mechanical coupling model.

By coupling the above results of the temperature field calculations at different speeds with the NPR spacer centrifugal force and the axial restraint force, Poisson’s ratios of the NPR spacer at different speeds before and after temperature consideration are obtained, which are shown in Figure 11. The results show that Poisson’s ratio of the NPR spacer, without considering the temperature, is only related to its structural parameters and does not vary with the speed. When the temperature is considered, the absolute value of Poisson’s ratio of the NPR spacer decreases with increasing speed; however, it is greater than that when the temperature is not considered. This is because the NPR spacer exhibits thermal elongation when the temperature is considered. At low speeds, both the centrifugal force and the radial strain are small. As the speed increases, although both the radial and axial strains increase, the rate of increase in the axial strain is less than that in the radial strain.

The elongations of the NPR spacer before and after considering the temperature are shown in Figure 12. It can be found that the elongation of NPR spacer considering temperature is greater than that without considering temperature at the same speed. Particularly, when the rotation speed is 2000 r/min, the elongations of the NPR spacer considering temperature and without considering temperature are 18.2 μm and 1.2 μm, respectively, and the difference between them is 0.0170 mm. When the speeds are 4000 r/min, 6000 r/min, and 8000 r/min the differences of axial elongation of NPR spacer considering and not considering the temperature are 20.3 μm, 21.3 μm, 23.0 μm, and 27.8 μm respectively. This is due to the fact that when temperature is considered, the axial elongation of NPR spacer includes the axial elongation caused by centrifugal forces and the thermal elongation caused by temperature change. The temperature of the spacer increases with the speed, while the thermal elongation of the spacer increases with the temperature. When the speed was increased from 2000 r/min to 8000 r/min, the elongation of NPR spacer with and without the consideration of temperature changed by 28.5 μm and 17.7 μm, respectively. This is due to the increase in thermal elongation caused by the increase in spacer temperature as the speed rises from 2000 r/min to 8000 r/min.

Figure 13 shows the output forces of the NPR spacer at different speeds before and after considering the temperature. When the speed increases from 2000 r/min to 8000 r/min, the output force of the NPR spacer considering (without considering) the temperature increases from 591.4 N (52.1 N) to 2200.1 N (783.4 N) and the output force of NPR spacer with and without the consideration of temperature changed by 731.3 N and 1608.1 N. At the same speed, the NPR spacer output force is greater when the temperature is considered than when it is not considered. When the speed is 2000 r/min, 4000 r/min, 6000 r/min, and 8000 r/min the differences of the NPR spacer output forces considering and not considering the temperature are 539.3 N, 784.1 N, 898.2 N, 1087.6 N, and 1416.7 N, respectively. The effect of speed on the elongation of the NPR spacer is less than that of temperature. This is because the temperature affects the thermal elongation of the NPR spacer, which is much larger than the axial elongation of the NPR spacer caused by the speed effect. The output force of the spacer is positively related to its axial elongation.

In summary, the temperature affects the Poisson’s ratio value, elongation, and output force of the NPR spacer; thus, when an NPR spacer operates in an environment with temperature change, the temperature effect cannot be neglected when designing its structure. In order to further identify the effects of temperature and rotational speed on the axial displacement and output force of the NPR spacer, the effects of different speeds and temperatures on the axial elongation and output force of the NPR spacer were subsequently analyzed by both simulation and experiments.

### 4.4. Discussion of Simulation and Experimental Results

In order to verify the force output characteristics of the NPR spacer under actual working conditions, an NPR spacer thermo-mechanical coupling test rig was set up, as shown in Figure 14. The test rig consists of a driving motorized spindle, loading device, measuring instrument, specimen, and computer. In the experiments, a supporting shaft is driven by the motorized spindle and an experimental shaft is connected to the supporting shaft by a BT40 tool-holder. The spindle speed is adjusted by an inverter (model VFD075V46A2). An HBM-U3 force sensor is employed, possessing a measurement range of −5 kN–5 kN and an accuracy of 0.5 N. A KF2306-5SUM eddy current displacement sensor is used and possesses a measurement range of 500 μm and accuracy of 0.1 μm. A Fluke-561 temperature sensor is employed and possesses a measuring range of −40 °C–550 °C and an accuracy of 0.1 °C. The signal acquisition device is Pak mobile MK Ⅱ. In the experiments, a loading device is used to apply an initial preload force of 1500 N on the bearing specimen at rest and the spindle speed is adjusted to a specific speed using a frequency converter. Subsequently, the NPR spacer is heated uniformly by a handheld heater and the temperature is measured using the temperature sensor. When the temperature reaches a stable value, the axial elongation and output force of the spacer are measured by the displacement and force sensors mounted on the bearing end caps, respectively.

The axial elongation and output force of the NPR spacer at 20 °C, 30 °C, 40 °C, 50 °C, and 60 °C were measured, respectively, at the set spindle speed. Each group of experimental conditions were measured three times and then the average value and standard deviation of the axial elongation and output force of the spacer were obtained. The experimental results are described by the error bar. The center of the error bars is the mean value of the experimental results and the upper and lower deviations correspond to the standard deviation of the experimental results. In order to describe the experimental results more clearly, the standard deviation of the axial elongation of the spacer is magnified by 5 times and the standard deviation of the output force is magnified by 10 times.

The effect of speed and temperature on the axial elongation of the NPR spacer is shown in Figure 15. It can be found that the maximum standard deviation of all experimental results is 0.9 μm. This shows that the performance of the NPR spacer is stable and that the experimental repeatability is good. At each temperature, the maximum deviation between the mean of the experimental results and simulation results is 1.4 μm. It shows that the simulation results match with the experimental results and proves the accuracy of the above method. Analysis of the effects of temperature and speed on spacer elongation using experimental data showed that the axial elongation of the NPR spacer increases with the increase in the temperature at a constant speed and, at a constant temperature, the axial elongation of the NPR spacer increases with the increase in the speed. At the same speed, the axial elongation of the spacer is almost equal with each 10 °C degree increase in temperature. Moreover, the temperature of the spacer is increased by 10 °C and the axial elongation of the spacer is greater than the axial elongation when the speed is increased by 1000 r/min. This is because the thermal elongation of the NPR spacer is related to the coefficient of linear expansion and the value of temperature change and its own length. The spacer is subjected to the same centrifugal force at the same speed and the axial elongation of the spacer caused by the centrifugal force is the same. Without considering other factors, the axial elongation of the spacer caused by the same temperature change at the same speed is equal. The effect of spacer speed on axial elongation is related to the size of the band gap in the spacer and the Poisson’s ratio of the spacer.

The effect of speed and temperature on the output force of the NPR spacer is shown in Figure 16. It can be found that the maximum standard deviation of all experimental results is 58 N. The maximum deviation between the experimental results and the average of the simulated results of the spacer output force at each temperature is 76 N. It shows that the simulation results match with the experimental results and proves the accuracy of the spacer thermo-mechanical coupling model. The NPR spacer axial output force increases as the temperature increases when the speed is constant and the NPR spacer output force increases as the speed increases when the temperature is constant. The NPR spacer output force, when the speed is increased from 2000 r/min to 6000 r/min at a specific temperature, is smaller than when this temperature is increased by 10 °C at 2000 r/min. For example, when the speed is 2000 r/min and the temperature is increased from 20 °C to 30 °C, the NPR spacer output force increases by 465 N. In comparison, when the temperature is 20 °C and the speed is increased from 2000 r/min to 6000 r/min, the NPR spacer axial elongation increases by 324 N. Temperature has a greater effect on the output force of the NPR spacer than speed. This is because when the temperature of the spacer is increased by 10 °C, the axial elongation of the spacer is greater than the axial elongation when the speed is increased by 1000 r/min and there is a positive correlation between the spacer output force and the axial displacement of the spacer.

## 5. Conclusions

In this paper, a novel method for bearing preload adjustment with low-porosity 3D NPR spacer was proposed. The deformation law and mechanical properties of the 3D NPR spacer were analyzed by both theoretical and experimental methods. The main conclusions are as follows:
(1)Based on the auxetic behavior of the NPR structure, an adaptive adjustment method for bearing preload was proposed. The results show that the 3D NPR spacer with concave hexagonal cell structure has better expansion performance.(2)Both the deformation characteristics and output force of the 3D NPR structure were discussed in consideration of the effect of heat and centrifugal force, which provides the theoretical basis for variable preload regulation of the bearing. It is found that the thermal effect has relatively larger influences on the axial elongation and output forces of the NPR spacer increase.(3)Experimental verification was carried out considering the influence of heat, centrifugal force, and axial restraint force. The results show that the standard deviation of output force and axial displacement of NPR spacer is slight. The small deviation between the experimental and simulated values of the output force and axial elongation indicates the possibility of engineering applications utilizing the proposed method.

## Figures and Tables

**Figure 1 materials-14-03421-f001:**
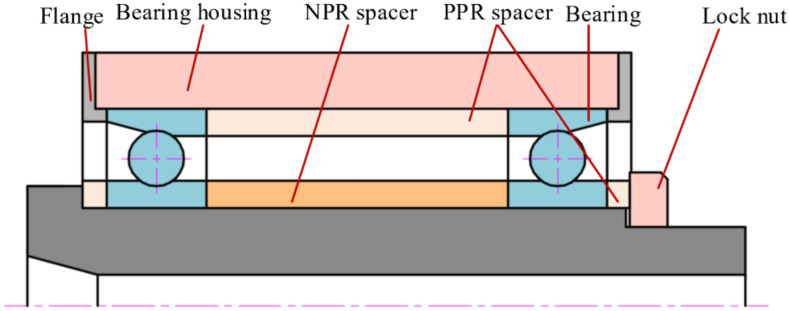
Schematic of spindle structure.

**Figure 2 materials-14-03421-f002:**
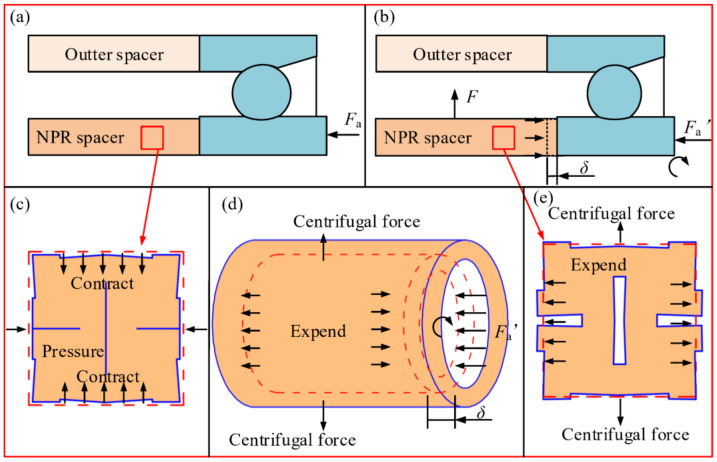
Principle of preload adjustment of NPR spacer. (**a**) Initial assembly stage of spacer, (**b**) Operation stage of spacer, (**c**) Cell contraction under compression, (**d**) Spacer expansion by centrifugal force, (**e**) Cell expansion under centrifugal force.

**Figure 3 materials-14-03421-f003:**
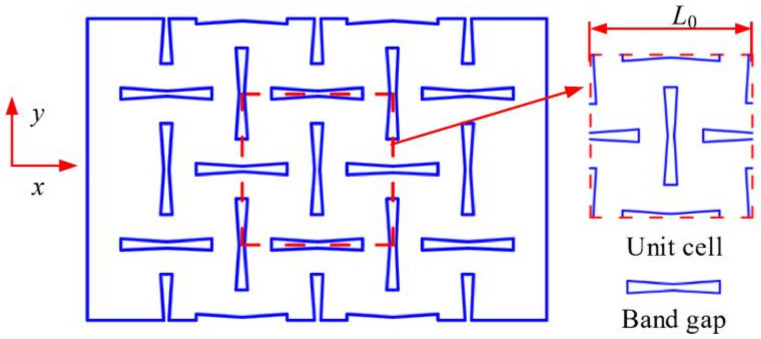
Periodic arrangement of cell structure.

**Figure 4 materials-14-03421-f004:**
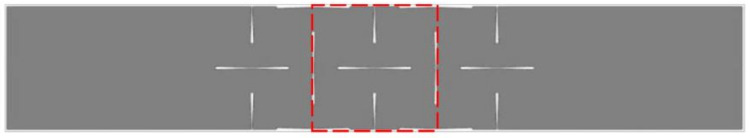
Finite element method model.

**Figure 5 materials-14-03421-f005:**
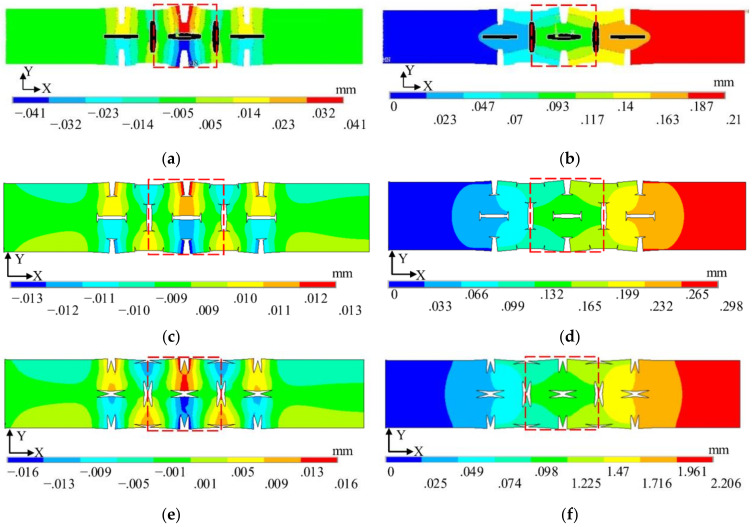
Contour maps of horizontal (*u*_x_) and vertical (*u*_y_) components of displacement field in different cells: (**a**) X-direction displacement of internally concave hexagonal, (**b**) Y-direction displacement of internally concave hexagonal, (**c**) X-direction displacement of H-shaped, (**d**) Y-direction displacement of H-shaped, (**e**) X-direction displacement of star-shaped, and (**f**) Y-direction displacement of star-shaped NPR periodic cell structures.

**Figure 6 materials-14-03421-f006:**
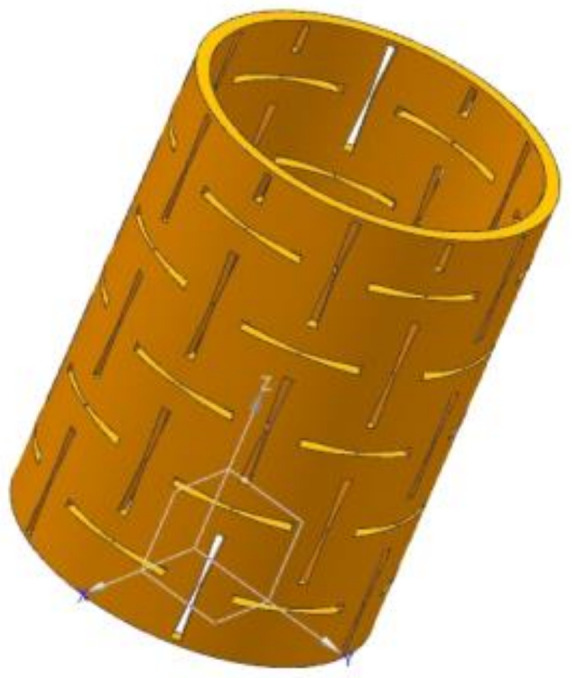
Three-dimensional NPR spacer structure.

**Figure 7 materials-14-03421-f007:**
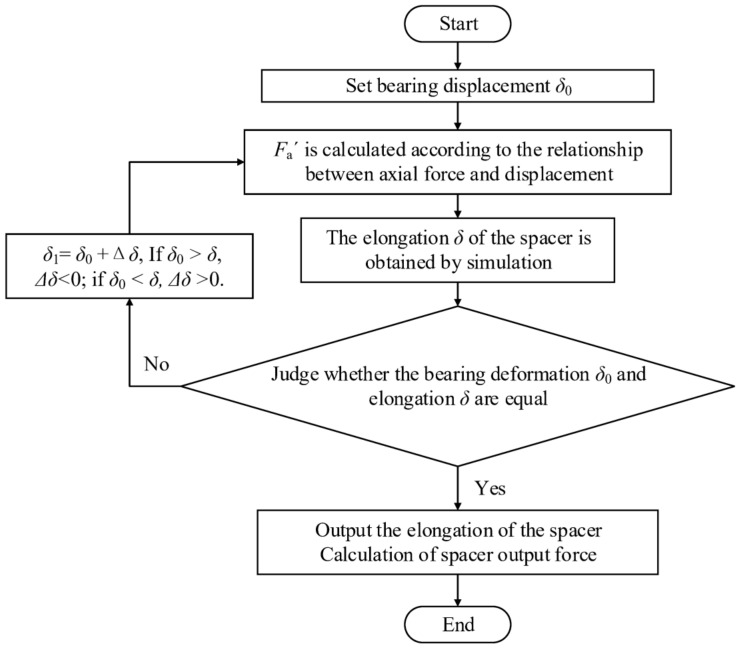
Flowchart of output force calculation.

**Figure 8 materials-14-03421-f008:**
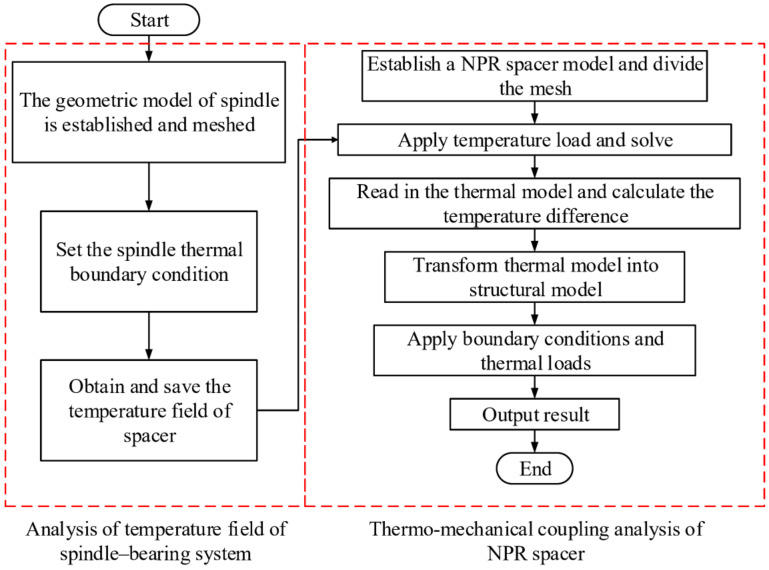
Thermo-mechanical coupling analysis flowchart.

**Figure 9 materials-14-03421-f009:**
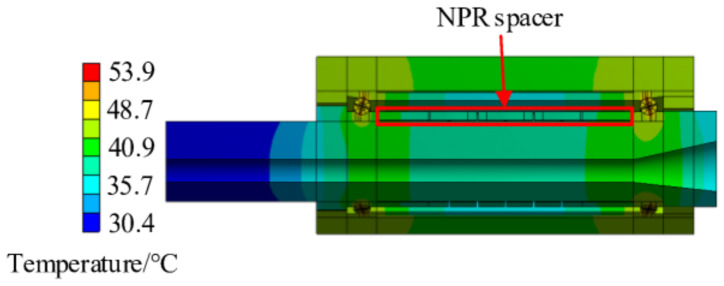
Spindle-bearing system temperature field at 8000 r/min.

**Figure 10 materials-14-03421-f010:**
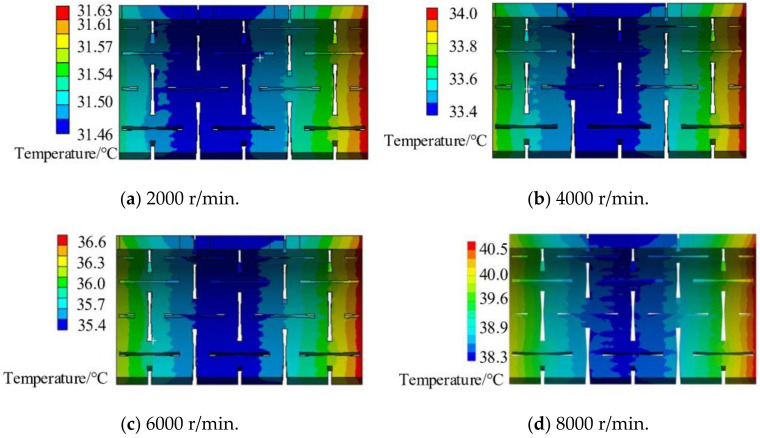
Temperature fields of NPR spacer at different speeds. (**a**) Temperature field of NPR spacer with speed of 2000 r/min, (**b**) Temperature field of NPR spacer with speed of 4000 r/min, (**c**) Temperature field of NPR spacer with speed of 6000 r/min, (**d**) Temperature field of NPR spacer with speed of 8000 r/min.

**Figure 11 materials-14-03421-f011:**
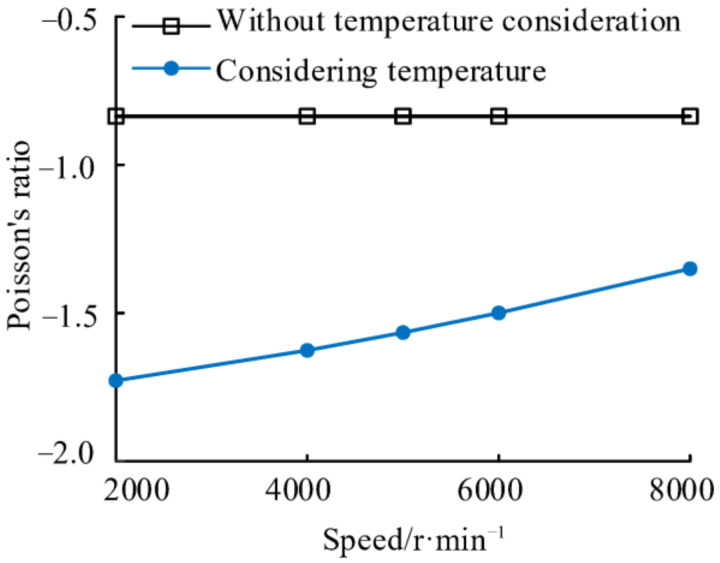
Poisson’s ratio variation of NPR spacer with speed.

**Figure 12 materials-14-03421-f012:**
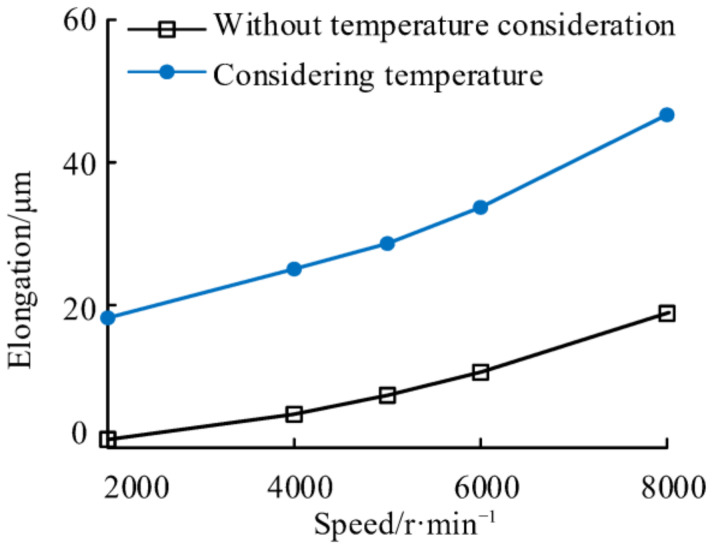
Elongations of NPR spacer at different speeds.

**Figure 13 materials-14-03421-f013:**
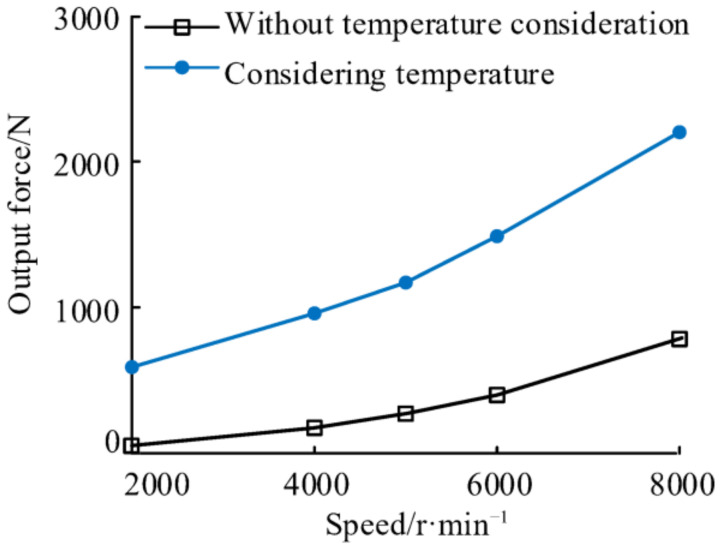
Output forces of NPR spacer at different speeds.

**Figure 14 materials-14-03421-f014:**
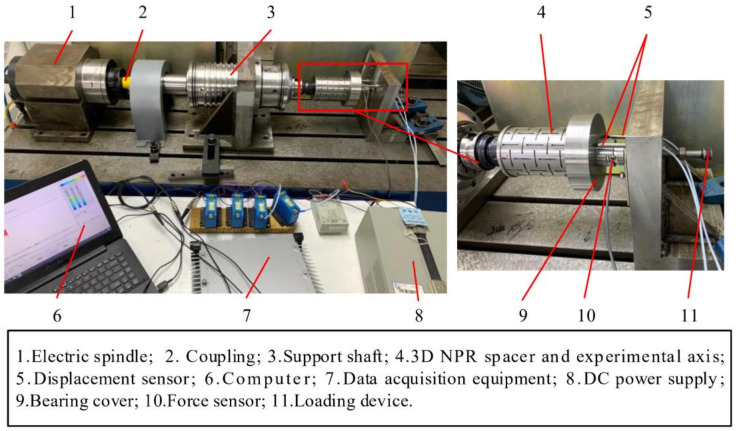
NPR spacer thermo-mechanical coupling test rig.

**Figure 15 materials-14-03421-f015:**
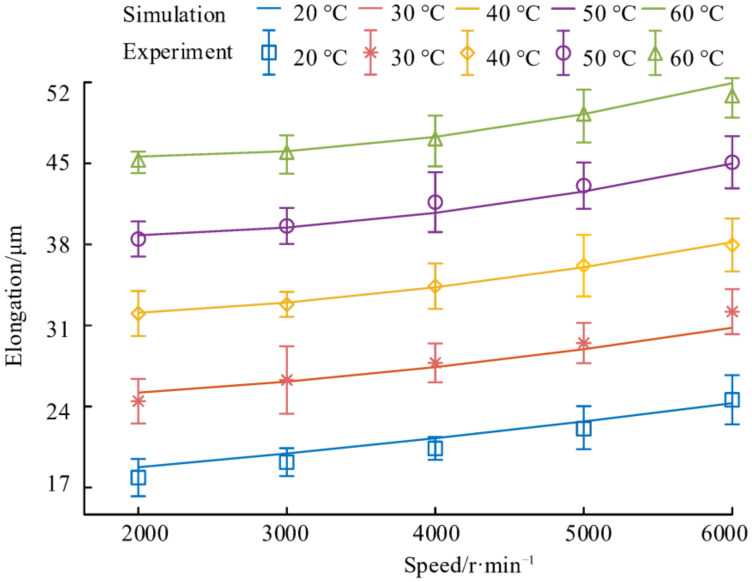
Effect of speed and temperature on the elongation of the NPR spacer.

**Figure 16 materials-14-03421-f016:**
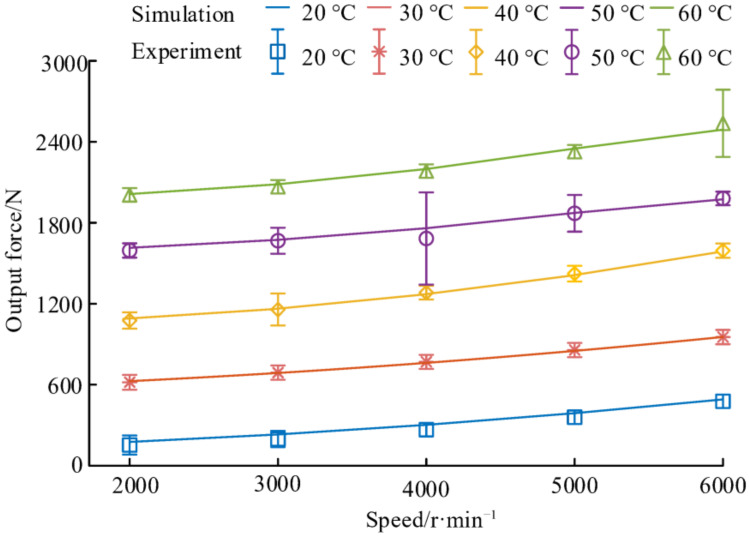
Effect of speed and temperature on the output force of the NPR spacer.

**Table 1 materials-14-03421-t001:** Band gap types and parameters.

	Type 1	Type 2	Type 3
Band gaps	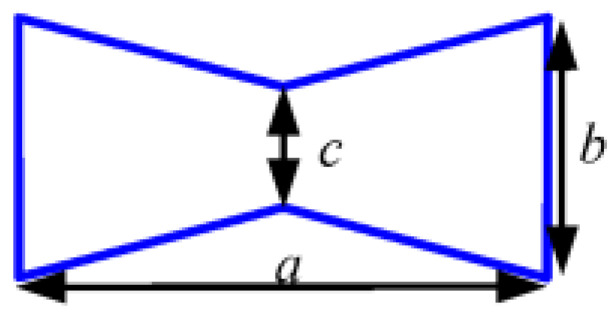	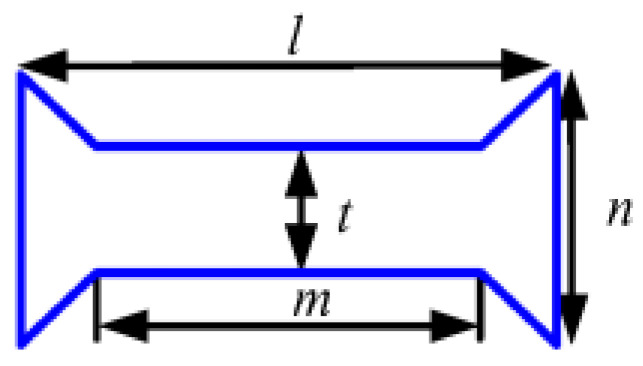	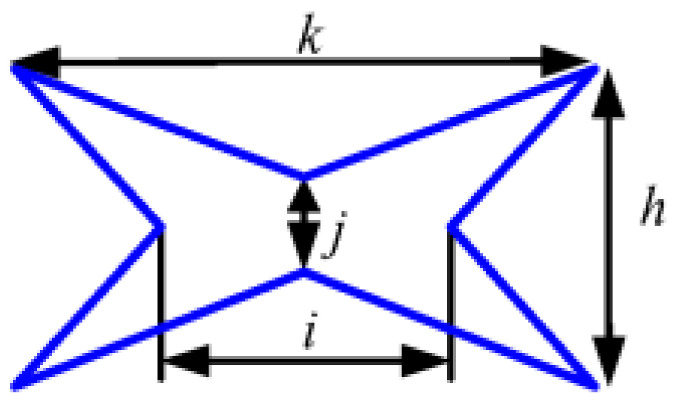
Structural parameters	*a* = 20 mm *b* = 14 mm *c* = 0.6 mm	*l* = 20 mm *m* = 18 mm *n* = 3.5 mm *t* = 1.5 mm	*k* = 20 mm *i* = 10 mm *h* = 4 mm *j* = 2 mm

## Data Availability

Detailed data are contained within the article.

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
