# Peer review of "Negative Poisson’s Ratio-Spacer Design and Its Thermo-Mechanical Coupling Analysis Considering Specific Force Output"

_materials, 2021, doi:10.3390/ma14123421_

Round 1
Reviewer 1 Report
The paper concerns the investigation of explores a low-porosity topological 3D NPR structure with high stiffness, considering thermo-mechanical coupling properties. The scientific results are many, but the work need a revision to better clarify the interpretation of the data and to better present the results. The discussion must be also improved and elaborated.
- The author should read the manuscript and clarify the command clearly. The authors have to care about typos. There are careless mistakes in some places.
- The authors should more describe the comparative analysis of several cell structures to preferably select the cell structure with the highest deformation capacity.
- Lines approx. 137-145 the authors describe the principle of the NPR spacer, but it is not clear. Pleaserestructured the sentences.
- In Figure 16, the authors motioned that the temperature has a greater impact on the axial elongation and output force of the NPR spacer more than the speed, please explain better how the temperature influenced.
The reviewer commands the paper to be accepted after minor revision.
Reviewer 2 Report
a deeper description on measured data processing formula and procedures is required
Conclusions are a repetition of the results and then a better analysis is required

Round 2
Reviewer 2 Report
the paper was correctly revised
Thanks